# Association analysis of Mycoplasma pneumoniae 23S rRNA gene mutation with refractory Mycoplasma pneumoniae pneumonia in children

Wen Li[1⊛], Wei Gao[2⊛], Xiaoyu Xiong[1], Xuan Tang[1], Aimin Li[1*]

**1** Department of Pediatrics, Jingzhou Central Hospital, Jingzhou Hospital Affiliated to Yangtze University, Jingzhou, China, **2** Department of Pediatrics, Xianning Central Hospital, The First Affiliated Hospital of Hubei University of Science and Technology, Xianning, China

⊛ These authors contributed equally to this study.
* liaimin0927@163.com

## Abstract

### Objective

This study investigates the current status of macrolide resistance in *Mycoplasma pneumoniae (MP)* and analyzes the relationship between mutations at the 23S rRNA A2063G and/or A2064G loci and refractory *Mycoplasma pneumoniae* pneumonia (RMPP).

### Methods

A retrospective analysis was conducted on 205 hospitalized children diagnosed with MPP at the Third Ward of the department of paediatrics, Jingzhou Central Hospital, from October 2023 to October 2024. Diagnosis was confirmed by pharyngeal *MP* nucleic acid testing and *MP* antibody titers (*MP*-Ab ≥ 1:160). All patients were screened for macrolide-resistant gene mutations (A2063G/A2064G). Patients were categorized based on the presence of macrolide-resistant gene mutations and RMPP diagnosis. Clinical features, laboratory results, and treatments were compared between groups. Multiple logistic regression was used to identify independent risk factors for RMPP.

### Results

Among 205 children with MPP, 157 (76.6%) harbored A2063G/A2064G mutations. Of the 77 children with RMPP, 71 (92.2%) carried these mutations, showing a significant association (*P* < 0.05). Compared to the non-resistant group, children with resistant mutations had prolonged fever duration, delayed defervescence after azithromycin, and required more bronchoscopic alveolar lavage (*P* < 0.05). The RMPP group exhibited more severe symptoms, longer fever and hospital stays, higher inflammatory markers (CRP, LDH, D-dimer, ESR, PCT; *P* < 0.001), and more frequent pulmonary

**Data availability statement:** All data are in the manuscript.

**Funding:** This research was supported by the Jingzhou Science and Technology Plan Project (No. 2024HD47); the funders had no role in study design, data analysis, or decision to publish.

**Competing interests:** The authors have declared that no competing interests exist.

consolidation, pleural effusion, plastic bronchitis, and extrapulmonary involvement. Multivariate analysis identified macrolide-resistant mutations, fever duration, and D-dimer level as independent risk factors for RMPP.

## Conclusion

The widespread macrolide resistance in *M. pneumoniae* (76.6% in this cohort) was associated with point mutations at the A2063G and/or A2064G loci in the 23S rRNA gene. The development of RMPP is linked to macrolide-resistant mutations, duration of fever and D-dimer levels. D-dimer emerged as the most predictive risk factor.

## 1. Introduction

*Mycoplasma pneumoniae (MP)* is a leading pathogen of pneumonia in China, with detection rates ranging from 6% to 80% [1]. *MP* pneumonia (MPP) is the most common type of community-acquired pneumonia (CAP) in children aged 5 years and older. While most cases of MPP are mild and can be effectively managed with a favorable prognosis, a small proportion of children experience rapid disease progression, resulting in severe or refractory cases [2,3]. Refractory MPP (RMPP) is defined as persistent fever, worsening clinical symptoms, and deteriorating lung imaging in children with MPP who have been treated with macrolide antibiotics for 7 days or more, along with the development of extrapulmonary complications [4,5]. RMPP is associated with significant intrapulmonary and extrapulmonary complications, which are protracted, costly, and often difficult to treat, leading to severe physical and psychological consequences for affected children [6,7].

MP, a pathogenic microorganism that exists independently of both viruses and bacteria, is widely distributed in nature. Due to its unique terminal structure, *MP* can adhere to respiratory mucosal epithelial cells, resulting in lung infections [8]. *MP* is inherently resistant to β-lactam antibiotics due to the absence of a cell wall, while aminoglycosides pose risks of ototoxicity and nephrotoxicity. Tetracyclines are generally avoided in children under 8 years of age. Macrolide antibiotics, such as azithromycin (15-membered ring) and erythromycin (14-membered ring), remain the first-line treatment for MPP in pediatric patients [9–11]. The 23S rRNA, located on the 50S large subunit of the ribosome, plays a central role in the ribosome's catalytic activity. The V region of the 23S rRNA contains peptidyl transferase enzyme activity, with the P site of the ribosomal 23S rRNA being the target of macrolide antibiotics. These antibiotics inhibit protein synthesis by blocking peptide chain elongation, thereby exerting their antimicrobial effects [12]. Mutations in this region can reduce the affinity of macrolides for the ribosome, leading to drug resistance in *MP* [13–15]. Over the past few years, the global prevalence of macrolide-resistant *MP* has risen steadily due to widespread antibiotic use and environmental factors [16–18]. A study by Tanaka et al. [19] on *MP* infections in Japanese children from 2008 to 2015 found macrolide-resistant *MP* to be the most prevalent strain. Similarly, Akashi et al. [20] reported 221 cases of macrolide-resistant mutations (MRM) among 383

MP infections, with a resistance rate of 57.7%. These mutations primarily occurred at the A2063 or A2064 sites in the V region of the 23S rRNA. Zheng et al. [21], through a study across six research institutions in the USA from August 2012 to April 2014, confirmed the presence of macrolide-resistant *MP* through PCR detection of mutations at the 23S rRNA locus, specifically the A2063G mutation. In China, various studies have identified resistance mutations, particularly at the A2063A→G and A2064A→G loci, with less frequent mutations at the 2067 and 2617 loci [13,22,23]. Mutations at positions A2063 and A2064, specifically A→G, have been strongly associated with high levels of macrolide resistance [24,25]. Additionally, mutations in structural domains II and V of ribosomal proteins L4 and L22 of *MP*, though less frequently reported, have also been explored as potential mechanisms of resistance [26,27].

In recent years, the increasing frequency of macrolide use, coupled with the rise in drug resistance, has led to a higher detection rate of *MP* resistance to macrolides. In some epidemic years, particularly in Asia and China, the resistance rate has exceeded 90% [28–30]. Numerous studies have demonstrated that the prevalence of macrolide-resistant Mycoplasma is directly linked to a marked increase in severe or refractory MPP, with the most severe cases potentially leading to sequelae such as bronchiectasis, pulmonary atelectasis (PA), and obstructive bronchiectasis (OB) [30–32]. The widespread outbreaks of *MP* across China in the autumn and winter of 2023, characterized by high incidence, significant macrolide resistance, rapid progression, and increased plastic bronchitis (PB), posed substantial clinical challenges. This study aimed to investigate the association between macrolide-resistant gene mutations (A2063G/A2064G) and RMPP in children, identify independent risk factors, and evaluate their predictive value. Our goal is to facilitate early diagnosis and intervention to prevent the progression to RMPP and its serious complications.

## 2. Materials and methods

### 2.1. Research population

A cohort of 205 hospitalized children diagnosed with MPP, confirmed through *MP* nucleic acid and serum antibody testing, were enrolled from the pediatric ward of Jingzhou Central Hospital between 01/10/2023 and 31/10/2024. Following admission, all patients were tested for macrolide-resistant mutations at the A2063G and/or A2064G loci in domain V of the 23S rRNA gene, using TaqMan probe-based real-time PCR.

All data were accessed for research purposes from 05/11/2024–25/12/2024.

### 2.2. Inclusion and exclusion criteria

Inclusion criteria: Age between 2 months and 15 years. MPP diagnostic criteria: both of the following must be met: (1) Lung imaging indicative of pneumonia; (2) Positive pharyngeal test for *MP*-DNA or serum IgM with *MP*-Ab titer ≥ 1:160 (PA method). RMPP diagnostic criteria, as per the 2023 National Healthcare Commission guidelines: children with MPP who, after at least 7 days of appropriate macrolide therapy, still exhibit persistent fever, worsening clinical and radiological signs, along with the emergence of extrapulmonary complications.

Exclusion criteria: (1) Presence of bronchial foreign body, underlying lung disease, congenital heart disease, or congenital immunodeficiency; (2) Coexistence of systemic diseases such as leukemia, epilepsy, or nephrotic syndrome; (3) Incomplete clinical data or non-cooperation during treatment, including requests for early discharge.

### 2.3. Data collection

(1) General Information: Age, sex, duration of fever, duration of azithromycin treatment, duration of fever resolution after azithromycin, and length of hospitalization; (2) Laboratory tests: *MP*-DNA, A2063G and/or A2064G mutations, *MP*-IgM and Ab titers, white blood cell count (WBC), C-reactive protein (CRP), procalcitonin (PCT), erythrocyte sedimentation rate (ESR), lactate dehydrogenase (LDH), liver and kidney function, cardiac enzyme, random glucose, D-dimer, nucleic acids for seven respiratory pathogens (rhinovirus, adenovirus, respiratory syncytial virus,

respiratory syncytial virus, influenza A/B, human metapneumovirus, *MP*), sputum culture, and urine routine tests; (3) Imaging findings: Pulmonary solidity, extent of lung lesions, pleural effusion(PE); (4) Bronchoscopic interventions: Alveolar lavage, plastic bronchitis(PB); (5) Extrapulmonary systemic involvement; (6) Treatment: Use of antibiotics and/or glucocorticoids.

## 2.4. Grouping methodology

Patients were stratified into two comparison pairs: (1) macrolide-resistant vs. non-resistant groups, based on the detection of A2063G/A2064G mutations; and (2) RMPP vs. non-RMPP groups, according to the standard diagnostic criteria for refractory disease.

## 2.5. Experimental methods and results analysis

### 2.5.1. Experimental methods.
The detection was performed using the commercial Mycoplasma pneumoniae Nucleic Acid and Antibiotic Resistance Mutation Site Detection Kit (Fluorescent PCR Method) (Jiangsu Molabio Biotechnology Co., Ltd., China) following the manufacturer's instructions. Briefly, the primers and TaqMan probes were designed to target the conserved region of the P1 gene (VIC-labeled) and the mutation sites A2063G and A2064G in domain V of the 23S rRNA gene (FAM-labeled). Each run included no-template controls (NTCs) and the kit-provided positive controls (weak and strong) for quality assurance. The mutation detection was based on the cycle threshold (Ct) values as described in the results analysis section (2.5.2), with all controls performing within the expected ranges.

Primary Equipment: Shengxiang MA-6000 Real-Time Quantitative PCR System Shengxiang Biotechnology Co., Ltd., China

Primary Reagents: Mycoplasma pneumoniae Nucleic Acid and Antibiotic Resistance Mutation Site Detection Kit (Fluorescent PCR Method) Jiangsu Molabio Biotechnology Co., Ltd., China.

2.5.1.1 Sample Processing for Mycoplasma pneumoniae Nucleic Acid and Antibiotic Resistance Mutation Site Detection (Throat Swab).

2.5.1.1.1 Manual Nucleic Acid Extraction Method for Throat Swabs: Add 1 mL of physiological saline to the dry swab sample. Vortex to mix thoroughly, then aliquot the sample into 1.5 mL sterile centrifuge tubes. Transfer 50 µL to 1 mL of the sample into a centrifuge tube. Label the sample tube, place it in a centrifuge, and centrifuge at 12,000 rpm for 5 minutes. Discard the supernatant and retain the pellet at the bottom. Add 50 µL lysis buffer and 5 µL internal standard solution. Vigorously mix by shaking for 15 seconds using a shaker. Centrifuge briefly for a few seconds. Incubate at 95°C for 18 minutes. Allow to cool to room temperature. Centrifuge at 12,000 rpm for 5 minutes. Set aside for later use

2.5.1.2 Handling of Mycoplasma pneumoniae Nucleic Acid and Antibiotic Resistance Mutation Site Positive Reference Materials: Same as Sample Handling 2.5.1.1

2.5.1.3 Take several reaction tubes for Mycoplasma pneumoniae nucleic acid and drug resistance mutation site detection. The reaction system for each tube is as shown in the table below:

| Reagent | Volume |
|---|---|
| Buffer | 6uL |
| Primer/Probe | 2uL |
| Enzyme | 0.5uL |
| Water | 11.5uL |

2.5.1.4. FQ-PCR Detection Instrument Preparation (Amplification Zone): Power on and preheat the instrument. Set probe detection mode to: Reporter Dye 1: FAM; Reporter Dye 2: VIC; Reporter Dye 3: CY5.

Assay (amplification zone): Amplification detection. The cycling conditions for the Mycoplasma pneumoniae nucleic acid and drug resistance mutation site detection reagent on the MA-6000 real-time quantitative PCR instrument are set as follows: 50°C for 2 minutes, 95°C for 2 minutes; 91°C for 15 seconds; 64°C for 1 minute, repeated for 40 cycles; reaction volume is 25 μL. The fluorescence detection point is selected at 64°C for 1 minute.

### 2.5.2. Results analysis.

| Real-time PCR Results for Tested Samples | Identification Results |
|---|---|
| 2063 or 2064 Mutation Mycoplasma pneumoniae (FAM) (VIC) | |
| Ct ≥ 35.33 Ct < 35.01 | Mycoplasma pneumoniae positive, no A2063G and/or A2064G resistance mutations detected |
| Ct < 35.33 Ct < 35.01 | Mycoplasma pneumoniae positive, with A2063G and/or A2064G resistance mutations present |

## 2.6. Statistical methods

Statistical analysis was performed using SPSS 29.0 software. Categorical data were expressed as percentages (%), and group differences were assessed using the chi-square test ($\chi^2$). The normality of distribution for continuous variables was assessed using the Shapiro-Wilk test. Continuous data with a normal distribution were presented as mean ± standard deviation ($\bar{x} \pm s$), and group comparisons were conducted using the independent samples t-test. Data with a non-normal distribution were presented as median (interquartile range) [M (P25, P75)], and intergroup comparisons were performed using the Mann-Whitney U test (Wilcoxon rank-sum test). Variables with a univariate $P$-value < 0.05 were included as candidates in a multivariate logistic regression analysis to identify independent risk factors. Receiver operating characteristic (ROC) curves were constructed to evaluate the predictive performance of significant factors, with the area under the curve (AUC), sensitivity, and specificity reported. A $P$-value < 0.05 was considered statistically significant.

## 2.7. Ethics approval and consent to participate

This study has been approved by the Ethics Committee of Jingzhou Central Hospital and the Ethics Committee of Jingzhou Hospital Affiliated to Yangtze University ((No. 2024-192-01). The research process strictly adheres to the ethical principles of the Declaration of Helsinki. Written informed consent was obtained from the legal guardians of all patients upon admission. We undertake to implement stringent confidentiality measures; all data underwent de-identification prior to analysis to ensure no specific individuals could be identified. This study fully complies with ethical standards to maximise the protection of patient privacy.

## 3. Results

### 3.1. General conditions

This retrospective study included 205 children with MPP hospitalized between October 2023 and October 2024. All patients met the diagnostic criteria and underwent macrolide-resistance gene testing. The overall macrolide-resistance mutation (A2063G/A2064G) rate was 76.6% (157/205). Based on clinical progression, 77 (37.6%) patients were classified as having RMPP.

### 3.2. Association between macrolide-resistant gene mutations and RMPP

The prevalence of A2063G/A2064G mutations was significantly higher in the RMPP group than in the non-RMPP group (92.2% [71/77] vs. 67.2% [86/128]; $\chi^2 = 16.784$, $P < 0.001$), confirming a strong association between these mutations and the development of RMPP.

### 3.3. Analysis of clinical characteristics

**3.3.1. Comparison of macrolide-resistant and non-macrolide-resistant groups.** The macrolide-resistant group (n = 157) had a significantly higher proportion of male patients (56.1% vs 22.9%, $P<0.001$) compared to the non-resistant group (n = 48), while the median age was comparable (~6 years, $P=0.703$).

Clinically, the resistant group exhibited a longer duration of fever and a delayed time to fever resolution after azithromycin initiation (both $P<0.05$, Table 1). No significant intergroup differences were observed in cough duration or length of hospitalization.

Among laboratory parameters, only D-dimer levels were significantly elevated in the resistant group ($P=0.004$). White blood cell count, CRP, LDH, ESR, and PCT did not differ between groups.

Imaging findings, including the presence of solid lung lesions, number of lobes involved, extent of lesions, and occurrence of pleural effusion, were also comparable between the two groups (all $P>0.05$, Table 3).

**3.3.2. Comparison of RMPP and non-RMPP.** The RMPP (n = 77) and non-RMPP (n = 128) groups were comparable in terms of sex distribution (male: 48.1% vs 48.4%) and median age ($P>0.05$, Table 2).

Clinically, the RMPP group had a significantly longer duration of fever, time to fever resolution after azithromycin, and length of hospitalization (all $P<0.001$). No significant differences were observed in cough duration or total azithromycin treatment days.

Laboratory analysis revealed markedly higher levels of CRP, LDH, D-dimer, ESR, and PCT in the RMPP group (all $P<0.001$). White blood cell counts did not differ significantly between groups.

Imaging findings demonstrated a higher prevalence of pulmonary consolidation (54.5% vs 8.6%, $P<0.001$) and pleural effusion (13.0% vs 0%, $P<0.001$) in the RMPP group (Table 3). The number of affected lobes and laterality of lung involvement were similar between the two groups.

**Table 1. Clinical characteristics of the macrolide-resistant and non-macrolide-resistant groups.**

| General information | Macrolide-resistant (n = 157) | Non-macrolide-resistant (n = 48) | X²/Z | P-value |
|---|---|---|---|---|
| Age (years) | 6.00 (4.25, 7.00) | 6.15 (5.00, 7.00) | −0.381 | 0.703 |
| Sex | | | | |
| male | 88 | 11 | 16.163 | <0.001 |
| female | 69 | 37 | | |
| Duration of fever (days) | 7.00 (5.00, 8.00) | 6.00 (2.25, 7.00) | −2.632 | 0.008 |
| Duration of cough (days) | 13.00 (11.00, 16.00) | 13.00 (11.00, 15.00) | −0.308 | 0.728 |
| Duration of azithromycin treatment (days) | 5.00 (3.00, 6.00) | 5.00 (5.00, 7.00) | −3.015 | 0.003 |
| Duration of fever resolution after azithromycin (days) | 5.00 (3.00, 7.00) | 2.00 (1.00, 4.00) | −4.797 | <0.01 |
| Length of hospitalization (days) | 7.00 (6.00, 9.00) | 6.00 (6.00, 8.00) | −1.887 | 0.059 |
| WBC(×10^9/L) | 6.85 (5.46, 8.85) | 7.66 (5.49, 9.62) | −1.415 | 0.157 |
| CRP (mg/L) | 10.01 (4.79, 18.13) | 10.86 (2.68, 28.12) | −0.407 | 0.684 |
| LDH (U/L) | 330.20 (286.45, 404.00) | 316.80 (279.00, 368.97) | −1.323 | 0.186 |
| D-dimer (ng/mL) | 221.00 (141.50, 329.00) | 140.50 (113.00, 237.50) | −2.903 | 0.004 |
| ESR (mm/h) | 26.00 (20.00, 36.00) | 24.00 (15.00, 31.00) | −1.945 | 0.052 |
| PCT (ng/mL) | 0.20(0.10, 0.30) | 0.20(0.10, 0.30) | −1.894 | 0.058 |

Data are presented as the median (25th-75th percentile).

WBC: white blood cell count; CRP: C-reactive protein; LDH: lactate dehydrogenase; ESR: erythrocyte sedimentation rate; PCT: procalcitonin.

**Table 2. Clinical characteristics of RMPP and non-RMPP.**

| General information | RMPP (n = 77) | Non-RMPP (n = 128) | X²/Z | P-value |
|---|---|---|---|---|
| Age (years) | 6.30(4.90, 7.50) | 5.95 (4.40, 7.00) | −1.923 | 0.055 |
| Sex | | | | |
| Male | 37 | 62 | 0.003 | 0.957 |
| Female | 40 | 66 | | |
| Duration of fever (days) | 8.00 (7.00, 10.00) | 5.00 (2.00, 7.00) | −8.738 | <0.001 |
| Duration of cough (days) | 13.00 (11.00, 16.00) | 13.00 (11.00, 15.00) | −0.778 | 0.437 |
| Duration of azithromycin treatment (days) | 5.00 (3.00, 7.00) | 5.00 (3.00, 6.00) | −1.802 | 0.072 |
| Duration of fever resolution after azithromycin (days) | 7.00 (5.00, 8.00) | 3.00 (1.00, 4.00) | −9.251 | <0.001 |
| Length of hospitalization (days) | 8.00 (6.50, 10.00) | 6.00 (6.00, 7.00) | −5.377 | <0.001 |
| WBC (×10^9/L) | 7.13 (5.51, 8.72) | 7.08 (5.46, 9.37) | −0.332 | 0.740 |
| CRP (mg/L) | 14.50 (7.25, 39.46) | 8.05 (3.29, 13.60) | −4.540 | <0.001 |
| LDH (U/L) | 365.40 (307.20, 469.00) | 310.25 (279.36, 358.08) | −4.470 | <0.001 |
| D-dimer (ng/mL) | 345.00 (261.50, 532.50) | 144.00 (115.00, 195.50) | −10.165 | <0.001 |
| ESR (mm/h) | 33.00(26.00, 45.00) | 21.00 (15.00, 28.75) | −7.378 | <0.001 |
| PCT (ng/mL) | 0.30 (0.10, 0.70) | 0.10 (0.10, 0.30) | −4.920 | <0.001 |

Data are presented as the median (25th-75th percentile).

RMPP: refractory Mycoplasma pneumoniae pneumonia.

**Table 3. Imaging characteristics of the RMPP and non-RMPP groups, the macrolide-resistant and non-macrolide-resistant groups.**

| | RMPP (n = 77) | Non-RMPP (n = 128) | P-value | Macrolide-resistant (n = 157) | Non- macrolide-resistant (n = 48) | P-value |
|---|---|---|---|---|---|---|
| Solid lung lesion | 42 (54.5%) | 11 (8.6%) | <0.001 | 43 (27.4%) | 10 (20.8%) | 0.364 |
| Involvement of lung lobes | | | | | | |
| Single-leaf | 12 | 31 | 0.141 | 30 | 13 | 0.235 |
| Multi-leaf | 65 | 97 | | 127 | 35 | |
| Extent of lung lesions | | | | | | |
| Unilateral | 42 | 35 | 0.339 | 80 | 23 | 0.713 |
| Bilateral | 61 | 67 | | 77 | 25 | |
| Pleural effusion | 10 (13.0%) | 0 | <0.001 | 7 (4.5%) | 3 (6.3%) | 0.903 |

### 3.4. Analysis of bronchoscopic interventions

**3.4.1. Comparison of endoscopic interventions between the macrolide-resistant and non-macrolide-resistant groups.** Bronchoscopic alveolar lavage was performed more frequently in the macrolide-resistant group (40.1% [63/157] vs. 20.8% [10/48]; P = 0.015, Table 4). Among these, eight children in the resistant group required multiple lavage sessions. The incidence of plastic bronchitis (PB) did not differ significantly between the groups (8.9% [14/157] vs. 4.2% [2/48]; P = 0.443).

**3.4.2. Comparison of endoscopic interventions between the RMPP and non-RMPP groups.** Both bronchoscopic alveolar lavage and PB were significantly more prevalent in the RMPP group than in the non-RMPP group (lavage: 75.3% [58/77] vs. 11.7% [15/128]; PB: 19.5% [15/77] vs. 0.8% [1/128]; both P < 0.001, Table 4).

**Table 4. Bronchoscopic interventions in RMPP versus non-RMPP and macrolide-resistant versus non-macrolide-resistant groups.**

| | RMPP (n = 77) | Non-RMPP (n = 128) | *P*-value | Macrolide-resistant (n = 157) | Non- macrolide-resistant (n = 48) | *P*-value |
|---|---|---|---|---|---|---|
| Alveolar lavage | 58 (75.3%) | 15 (11.7%) | <0.001 | 63 (40.1%) | 10 (20.8%) | 0.015 |
| Plastic bronchitis | 15 (19.5%) | 1 (0.8%) | <0.001 | 14 (8.9%) | 2 (4.2%) | 0.443 |

## 3.5. Analysis of mixed infections

The overall rate of mixed infections did not differ significantly between the macrolide-resistant and non-resistant groups (22.9% [36/157] vs. 29.2% [14/48]; $\chi^2 = 0.775$, $P = 0.379$), nor between the RMPP and non-RMPP groups (26.0% [20/77] vs. 23.4% [30/128]; $P = 0.682$) (Table 5).

Among all mixed infections (n = 50), viral co-infections were most common (66.0%), primarily with rhinovirus, followed by bacterial (24.0%, predominantly *Haemophilus influenzae*) and combined viral-bacterial infections (10.0%). The distribution of these infection types did not differ significantly between the comparison groups ($P > 0.05$).

## 3.6. Analysis of extrapulmonary complications

The incidence of extrapulmonary complications did not differ significantly between the macrolide-resistant and non-resistant groups (20.4% [32/157] vs. 22.9% [11/48]; $P = 0.706$, Table 6). In contrast, complications were markedly more frequent in the RMPP group than in the non-RMPP group (39.0% [30/77] vs. 10.2% [13/128]; $P < 0.001$).

**Table 5. Mixed infections in RMPP and non-RMPP groups, macrolide-resistant and non- macrolide-resistant groups.**

| | RMPP (n = 77) | Non-RMPP (n = 128) | *P*-value | Macrolide-resistant (n = 157) | Non- macrolide-resistant (n = 48) | *P*-value |
|---|---|---|---|---|---|---|
| Mixed infection | 20 (26.0%) | 30 (23.4%) | 0.682 | 36 (22.9%) | 14 (29.2%) | 0.379 |
| Combined viral infections | 14(70.0%) | 19(63.3%) | 0.904 | 22 (61.1%) | 11 (78.6%) | 0.624 |
| Combined bacterial infections | 4(20.0%) | 8(26.7%) | | 10 (27.8%) | 2 (14.3%) | |
| Multiple infection | 2(10.0%) | 3(10.0%) | | 4 (11.1%) | 1 (7.1%) | |

**Table 6. Extrapulmonary complications in RMPP versus non-RMPP and macrolide-resistant versus non-macrolide-resistant groups.**

| | | RMPP (n = 77) | Non-RMPP (n = 128) | *P*-value | Macrolide-resistant (n = 157) | Non- macrolide-resistant (n = 48) | *P*-value |
|---|---|---|---|---|---|---|---|
| Extrapulmonary systemic involvement | | 30 (39.0%) | 13(10.2%) | <0.001 | 32 (20.4%) | 11 (22.9%) | 0.706 |
| Digestive systems | Elevated ALT and/or AST | 15 | 2 | | 14 | 3 | |
| | Diarrhea | 1 | 1 | | 1 | 1 | |
| Skin and mucous membrane damage | Rash | 5 | 2 | | 7 | 0 | |
| Hematological system | Thrombocytopenia | 0 | 1 | | 1 | 0 | |
| Endocrine disorder | Hyperglycemia | 3 | 1 | | 4 | 0 | |
| Circulatory system | Elevated cardiac enzymes | 3 | 1 | | 2 | 2 | |
| Urinary system | Positive urine protein | 2 | 1 | | 1 | 2 | |
| | Positive urine occult blood | 1 | 4 | | 2 | 3 | |

RMPP: refractory Mycoplasma pneumoniae pneumonia; ALT: alanine aminotransferase; AST: aspartate aminotransferase.

Among the 43 total cases with complications, hepatic dysfunction was most common, followed by rash, diarrhea, elevated cardiac enzymes, hyperglycemia, and urinary abnormalities (proteinuria or occult blood). One case of hematological involvement (thrombocytopenia) was recorded.

### 3.7. Analysis of antimicrobial and hormonal therapy

The distribution of treatment regimens differed significantly between both comparison pairs (both $P<0.001$, Table 7). In the macrolide-resistant group, the most common regimen was a new tetracycline/levofloxacin combined with methylprednisolone (43.3%), followed by new tetracycline/levofloxacin alone (39.5%). In contrast, the non-resistant group most frequently received azithromycin alone (58.3%).

Similarly, the RMPP group was most often treated with a new tetracycline/levofloxacin plus methylprednisolone (68.8%), whereas the non-RMPP group most commonly received new tetracycline/levofloxacin alone (40.6%), followed by azithromycin alone (32.8%).

### 3.8. Multiple logistic regression analysis for factors predicting RMPP

Variables with a P-value $<0.05$ in the univariate comparisons between the RMPP and non-RMPP groups (as shown in Tables 2–4, 6,7) were considered candidates for the multivariate logistic regression model. These included: duration of fever, CRP, LDH, D-dimer, ESR, PCT, pulmonary consolidation, pleural effusion, bronchoscopic alveolar lavage, plastic bronchitis, extrapulmonary complications, and treatment regimen. To avoid overfitting and multicollinearity, we selected the most clinically relevant and statistically robust predictors (macrolide-resistant gene mutation, duration of fever, and D-dimer) for the final model. In the final model, each of these factors was independently associated with an increased risk of RMPP (all $P<0.05$, Table 8).

### 3.9. ROC curve analysis of RMPP

The predictive performance of the three independent risk factors was evaluated using Receiver operating characteristic (ROC) curve analysis (Fig 1, Table 9). D-dimer demonstrated the highest discriminative ability (AUC = 0.924, 95%

**Table 7. Antimicrobial and hormonal therapy in RMPP and non-RMPP groups, macrolide-resistant and non- macrolide-resistant groups.**

|  | RMPP (n=77) | non-RMPP (n=128) | P-value | Macrolide-resistant (n=157) | non- macrolide-resistant (n=48) | P-value |
|---|---|---|---|---|---|---|
| New tetracycline /levofloxacin + methylprednisolone | 53 (68.8%) | 20 (15.6%) | <0.001 | 68 (43.3%) | 5 (10.4%) | <0.001 |
| Azithromycin + methylprednisolone | 10 (13.0%) | 14 (10.9%) |  | 12 (7.6%) | 12 (25.0%) |  |
| Azithromycin | 1 (1.3%) | 42 (32.8%) |  | 15 (9.6%) | 28 (58.3%) |  |
| New tetracycline/ levofloxacin | 13 (16.9%) | 52 (40.6%) |  | 62 (39.5%) | 3 (6.3%) |  |

**Table 8. Multiple logistic regression analysis for the related factors predicting the RMPP.**

| Variable | B | SE | Wald | P-value | OR | 95% CI | |
|---|---|---|---|---|---|---|---|
|  |  |  |  |  |  | Lower | Upper |
| Macrolide-resistant gene mutation | 2.880 | 0.920 | 9.806 | 0.002 | 17.806 | 2.937 | 107.971 |
| Duration of fever | 0.601 | 0.148 | 16.591 | <0.001 | 1.824 | 1.366 | 2.436 |
| D-dimer | 0.022 | 0.004 | 30.144 | <0.001 | 1.022 | 1.014 | 1.030 |

SE: standard error; OR: odds ratio; 95% CI: 95% confidence interval.

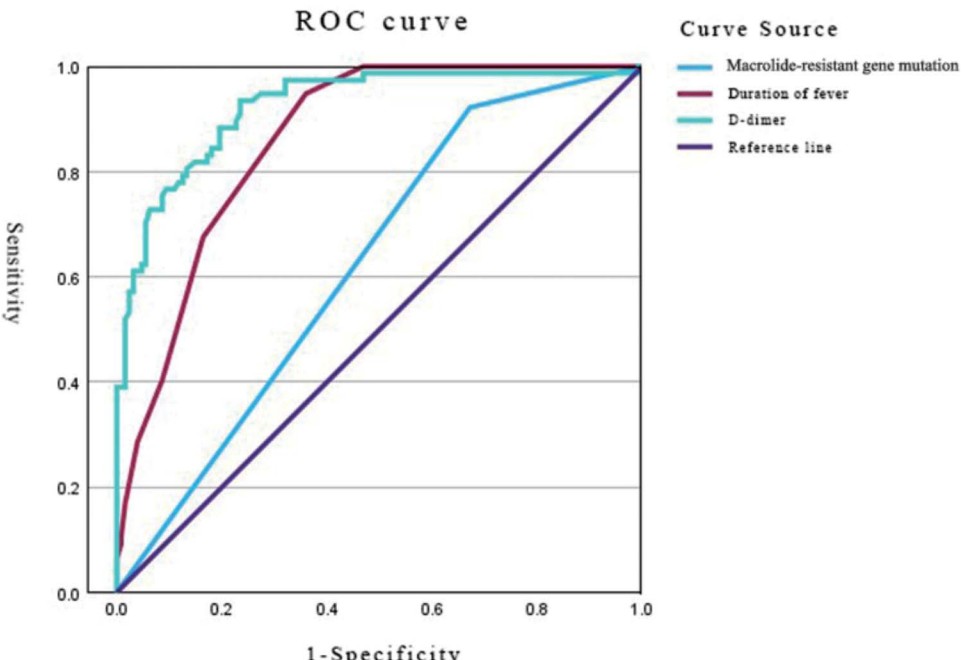

**Fig 1. ROC curves for predicting RMPP based on macrolide-resistant gene mutation, duration of fever, and D-dimer level. D-dimer showed the highest discriminatory power (AUC = 0.924, 95%CI: 0.886–0.962).**

**Table 9. Analysis of ROC curves for RMPP predictors.**

| Variable | AUC | Cut-off value | Sensitivity (%) | Specificity (%) | 95% CI | Youden index | P-value |
|---|---|---|---|---|---|---|---|
| Macrolide-resistant gene mutation | 0.625 | NA | 0.92 | 0.33 | 0.549~0.701 | 0.250 | 0.001 |
| Duration of fever | 0.862 | 6.50 | 0.95 | 0.64 | 0.813~0.910 | 0.589 | 0.000 |
| D-dimer | 0.924 | 201.50 | 0.94 | 0.77 | 0.886~0.962 | 0.701 | 0.000 |

ROC: receiver operating characteristic; AUC: area under the ROC curve; NA: not applicable; Cut-off value: the threshold that maximizes the Youden index (sensitivity + specificity − 1); 95% CI: 95% confidence interval; P-value: probability that the observed AUC differs from the null hypothesis value of 0.5 (no discrimination).

CI: 0.886–0.962), with an optimal cut-off of 201.50 ng/mL yielding a sensitivity of 94% and a specificity of 77% (Youden index = 0.701). Fever duration also showed substantial predictive value (AUC = 0.862, 95% CI: 0.813–0.910), whereas the macrolide-resistant gene mutation had limited utility (AUC = 0.625, 95% CI: 0.549–0.701).

## 4. Discussion

MPP is a prevalent cause of community-acquired pneumonia in children, with infection rates among those over 5 years old reaching up to 50% [33,34]. Recent reports have increasingly highlighted the association between mutations at loci 2063 and 2064 in the V region of the 23S rRNA structural domain and the development of RMPP, leading to MP drug resistance [22,35–37]. Most scholars believe that the pathogenesis of RMPP is closely linked to *MP* resistance to macrolides, delayed treatment, and excessive immuno inflammation. The emergence of macrolide-resistant *MP* (MRMP) severely diminishes the efficacy of macrolides, leading to disease progression or prolongation, ultimately triggering the development of RMPP [28,38].

In this study, among patients with RMPP, 71 cases had macrolide-resistant gene mutations while 6 did not, Statistical analysis confirmed a significant association between the presence of macrolide-resistant mutations and RMPP (OR: 17.8, 95% CI: 2.9–107.9). However, this association requires careful interpretation. First, while the odds ratio was high (OR: 17.8), its 95% confidence interval was notably wide (2.9–107.9), indicating a degree of imprecision in the estimate, which is likely influenced by the relatively small number of RMPP cases without the mutation (n = 6). This underscores the need for validation in larger cohorts. Furthermore, ROC curve analysis for the mutation yielded an AUC of only 0.625 (95% CI: 0.549–0.701), with a sensitivity of 92% but a specificity of 33% (Youden index = 0.250). These results suggest that while the mutation is strongly associated with RMPP, its standalone value for clinical prediction is limited. This limited predictive utility likely reflects the multifactorial pathogenesis of RMPP. Progression to refractory disease is not determined by macrolide resistance alone but involves a complex interplay of host and pathogen factors, including dysregulated immune responses, high bacterial load, hypercoagulability, mucus hypersecretion, and co-infections [7,39].

RMPP has been reported to be more prevalent in older age groups, particularly in school-age children over 5 years, likely due to the heightened immune response with increasing age, which results in more extensive immune-induced intrapulmonary and extrapulmonary damage [40]. However, this study found no statistically significant age differences between macrolide-resistant and non- macrolide-resistant groups or between the RMPP and non-RMPP groups.

Clinically, persistent fever is a hallmark of RMPP, primarily induced by *MP* infection through immune responses and the release of inflammatory mediators. *M. pneumoniae* damages respiratory epithelium through adhesion and toxin secretion, including CARDS toxin. This damage triggers the immune system to produce large amounts of inflammatory mediators like interleukins (IL) and tumor necrosis factor (TNF), which elevate body temperature. Moreover, due to its lack of a cell wall, MP is resistant to many antibiotics, and the emergence of resistant strains contributes to recurrent fever, or even hyperthermia [41,42]. Many studies have identified recurrent fever as a prominent feature of RMPP and a risk factor for its progression [43]. Gong et al. [44] found that fever lasting more than 10 days was an independent risk factor for RMPP. In this study, fever duration was identified as a significant predictor for RMPP. ROC analysis determined an optimal cut-off of >6.5 days (AUC = 0.862, sensitivity 95%, specificity 64%, Youden index = 0.589). Clinically, disease progression was commonly observed when fever persisted beyond 5–7 days. Therefore, a fever duration exceeding 6.5 days serves as a valuable and practical clinical warning sign, thus providing a crucial window for early intervention to prevent severe pulmonary and extrapulmonary complications.

D-dimer is a fibrin degradation product, and elevated levels of D-dimer indicate hypercoagulability and secondary hyperfibrinolysis, playing a pivotal role in the diagnosis, efficacy assessment, and prognosis of thrombotic diseases. Numerous studies have demonstrated a high predictive value of D-dimer for RMPP [6,34,45]. A retrospective study by Fu et al. [46] on thrombophilia complicating MP infection found that elevated D-dimer levels correlated with more severe disease. Among the 14 children with elevated D-dimer, three had cerebral embolism, resulting in varying degrees of sequelae. A study by Fengqin Liu et al. analyzed 30 children diagnosed with pediatric thrombotic pulmonary embolism (PET) at the Provincial Hospital of the First Medical University of Shandong from January 2017 to December 2021, 23 of whom had *MP* infections. Of these, 16 exhibited macrolide-resistant mutations at loci A2063G or A2064G, all of whom were diagnosed with RMPP. The results indicated that all these cases had significantly elevated D-dimer levels and required anticoagulant therapy [47]. In the current study, several of the 77 RMPP cases had D-dimer levels of 1147 ng/mL, 2852 ng/mL, 3393 ng/mL, 5122 ng/mL, and as high as 5555 ng/mL, while the normal range is 0–232 ng/mL. These RMPP cases, which were accompanied by PB or PE, all underwent bronchoscopic alveolar lavage, with some requiring multiple treatments. Fortunately, no serious complications, such as pulmonary embolism or other organ embolisms, occurred following active anti-infection and anticoagulation therapy.

D-dimer levels reflect the hypercoagulable state in the body, which is also believed to be one of the mechanisms contributing to RMPP. Zhu et al. [48] reported a rare case of a hereditary and acquired hypercoagulable state in an adolescent. The 11-year-old child, during RMPP treatment, experienced low back discomfort and was subsequently diagnosed

with renal vein thrombosis *via* computed tomography (CT). The child was ultimately diagnosed with acquired hereditary thrombophilia and antithrombin (AT) III deficiency based on family history. This case highlights the need to consider hereditary thrombophilia as a complication of thromboembolism during MP infection. The development of a hypercoagulable state due to the primary disease may also play a significant role in the progression of MPP to RMPP. In this study, children with RMPP and elevated D-dimer levels had a longer duration of fever, more severe clinical symptoms, and more prominent pulmonary imaging changes compared to non-RMPP children. Higher D-dimer levels were associated with an increased incidence of both pulmonary and extrapulmonary complications, such as PB, PE, and impaired liver and kidney function, as well as greater treatment difficulty. ROC curve analysis revealed a D-dimer AUC of 0.924, sensitivity of 94%, specificity of 77%, and a Youden index of 0.701, with a predictive cut-off value of 201.50 ng/mL. These findings demonstrate that D-dimer offers the highest predictive value and diagnostic accuracy for RMPP when compared to macrolide-resistant gene mutations and fever duration.

Compared to the non-resistant group, children with macrolide-resistant mutations had a longer duration of fever, a delayed response to azithromycin, and a longer hospital stay. They also exhibited significantly higher D-dimer levels, while other laboratory markers (WBC, CRP, LDH, PCT, ESR) showed no difference. Imaging findings—including consolidation, lobe involvement, lesion extent, and pleural effusion—were comparable between the groups. Although the resistant group required bronchoscopic alveolar lavage more frequently, the incidence of plastic bronchitis (PB), mixed infections, and extrapulmonary complications did not differ significantly. Zhou et al. [23] reported that macrolide-resistant MPP was associated with a higher incidence of both intrapulmonary and extrapulmonary complications, with 96% of children harboring the A2063G resistance mutation. Extrapulmonary complications were most commonly observed in the gastrointestinal system, followed by hepatic, cardiovascular, and cutaneous or mucosal damage. However, no significant difference in extrapulmonary complications was observed between the macrolide-resistant and non-resistant groups. This finding may be interpreted in light of our primary results. Extrapulmonary complications were strongly associated with disease severity, as defined by the RMPP phenotype (Table 6). Since the development of RMPP is multifactorial and not solely determined by macrolide resistance, the occurrence of such complications might be more directly linked to the severe/refractory disease state itself rather than to the resistance genotype alone. This interpretation is further supported by the comparable rates of other severity markers (e.g., imaging findings) between the resistance groups. Alternatively, the null finding could be due to the limited sample size inherent in this single-center retrospective design, which may lack the power to detect a modest association. In contrast, treatment regimens differed markedly between groups, with the resistant group most often receiving combination therapy (novel tetracyclines/levofloxacin plus methylprednisolone) or alternative antibiotics alone, whereas the non-resistant group was predominantly treated with azithromycin monotherapy. Macrolides have long been the first-line treatment for MPP, particularly in infants and schoolchildren, due to their efficacy and safety. However, the emergence and spread of macrolide-resistant strains have markedly reduced the efficacy of macrolides, making them less effective or even ineffective, leading to severe or refractory pneumonia [49,50] Among the recommended second-line treatments for MRMP, new tetracyclines, such as doxycycline and minocycline, are commonly used, followed by quinolone antibiotics like levofloxacin [51]. In a study by Okada et al. [52], which investigated a 2011 outbreak of macrolide-resistant *MP* infection in Japan, 87.1% of 202 MPP cases were caused by MRMP. Treatment with doxycycline or minocycline led to rapid defervescence and clinical improvement within 72 hours. However, tetracyclines can cause enamel hypoplasia and tooth discoloration and are therefore recommended only for children over 8 years of age. Quinolone antibiotics carry the risk of tendon rupture and are contraindicated for children under 18. As such, antimicrobial choice must be carefully considered, taking into account the child's condition, family consent, and adherence to recommended dosages [51,53].

In this study, of the 205 children, 77 were classified into the RMPP group and 128 into the non-RMPP group, with no significant differences in gender or age between the two groups. Clinically, the RMPP group exhibited longer fever duration, prolonged azithromycin use, extended time to remission post-azithromycin administration, and longer hospitalization compared to the non-RMPP group. Laboratory examination revealed significantly higher levels of CRP, LDH, D-dimer,

ESR, and PCT in the RMPP group. Shen et al. [54] found that the RMPP group showed more severe clinical signs, with significantly elevated inflammatory markers such as CRP, IL-6, LDH, and D-dimer compared to the common pneumonia group. Wen et al. [55] identified serum ferritin (SF), D-dimer, and CRP as independent risk factors for RMPP. Huang et al. [6] further concluded that CRP, LDH, D-dimer, and lung consolidation were independent risk factors for RMPP. Several studies have indicated that LDH is an independent risk factor for RMPP [56–58]. However, in this study, although the RMPP group showed a higher mean level of LDH, logistic regression analysis did not identify LDH as an independent predictor of RMPP, with D-dimer emerging as a more significant predictor. Further research is needed to clarify this finding.

Imaging analysis revealed that the RMPP group exhibited more severe lung involvement, with a higher likelihood of concomitant pulmonary solid changes and pleural effusions, consistent with findings from other studies. Wang et al. [43] reported that children with RMPP had longer fever durations and hospitalizations, along with more severe lung lesions and pleural effusions. A meta-analysis by Gong et al. [44] of 15 independent studies found that prolonged fever (more than 10 days), extrapulmonary complications, pleural effusion, and ≥2/3 lung lesion involvement on chest radiographs were significantly associated with RMPP. Similarly, this study concluded that RMPP was more frequently associated with bronchoscopic intervention, with a higher incidence of alveolar lavage and PB in the RMPP group compared to the non-RMPP group. Specifically, only one child in the non-RMPP group had concomitant PB, and 15 received alveolar lavage, whereas 15 of the 77 children in the RMPP group developed PB and 58 underwent alveolar lavage. Zhang et al. [59] identified persistent fever, extrapulmonary complications, and pleural effusion as risk factors for early detection of PB in RMPP cases. MPP complicated by PB is a key factor contributing to the difficulty in treating RMPP, with the formation of mucus plugs being one of the mechanisms driving RMPP. When large solid lung lesions and recurrent fever are observed in imaging, timely bronchoscopic intervention and alveolar lavage can be beneficial in treating RMPP and preventing severe sequelae [60,61].

In this study, no statistically significant difference was observed in the occurrence of concurrent mixed infections between the RMPP and non-RMPP groups. Some reports have indicated that the development of RMPP is associated with mutations in MP resistance genes and co-infections with adenovirus (ADV) [62]. However, no significant correlation between RMPP and concurrent mixed infections was found in this study, which may be attributed to several factors. First, autumn and winter are peak seasons for various viral infections, such as influenza virus, rhinovirus, and human parvovirus, and the MPP cases selected in this study coincided with the viral epidemic season, making viral co-infections more likely. Second, this study was conducted at a single center, which may introduce biases in the findings. Further analysis through multi-center, multi-regional studies is necessary to provide more robust conclusions.

In this study, extrapulmonary complications were found to occur more frequently and with greater severity in the RMPP group compared to the non-RMPP group. The most common extrapulmonary manifestation was hepatic dysfunction, followed by rash, diarrhea, elevated cardiac enzymes, hyperglycemia, and positive urinary occult blood or urinary proteins. Additionally, one case presented with thrombocytopenia in the hematological system. Several studies have highlighted that RMPP can be associated with various systemic complications involving the cardiovascular, digestive, skin and mucosal, nervous, hematological, and urinary systems. Although *MP* is rarely isolated from extrapulmonary tissues, extrapulmonary manifestations are believed to result from immune-mediated responses to the infection [63]. *MP*-induced rash and mucositis (MIRM) is a rare clinical condition. Chen and Li [64]reported 10 cases of MIRM in children, where most cases involved significantly elevated inflammatory markers such as ESR, CRP, LDH, and D-dimer. Early treatment with glucocorticoids and immunoglobulin notably improved the condition. *MP* infection is also associated with serious neurological complications, including aseptic meningitis, encephalitis, and Guillain-Barré Syndrome (GBS) [65,66]. Nonspecific arthritis is a rare extrapulmonary complication of *MP* infection, with *MP* arthritis (MPA) predominantly observed in individuals with humoral immunodeficiency [67]. A case of endocarditis due to MP infection was reported in a 15-year-old boy with recurrent fever, which was later diagnosed following a significant increase in *MP* titer detected through serum IgM and IgG antibodies, and confirmed by blood culture isolating a typical *MP* strain [68]. Although endocarditis caused by *MP*

infection remains rare, further studies are needed to verify its role in such conditions. Poddighe [69] reviewed literature on MP-related hepatitis (MPRH), noting that MP infection can lead to acute hepatitis in children, likely mediated by immune responses. Additionally, there have been reports of severe MPP complicated by acute interstitial nephritis with MP isolated from renal biopsies [70]. In summary, the occurrence of serious extrapulmonary complications due to MP infection requires further investigation. More clinical studies and tests are essential to better understand these manifestations. Early recognition and prevention of severe sequelae remain a critical task for pediatricians.

In this study, the distribution of therapeutic regimens between the RMPP and non-RMPP groups showed significant differences. The RMPP group was most commonly treated with novel tetracyclines or levofloxacin combined with methylprednisolone, whereas in the non-RMPP group, the most common treatment was azithromycin alone, followed by novel tetracyclines or levofloxacin alone. One of the key mechanisms behind the lung injury induced by RMPP is the body's abnormal immune response [8], and glucocorticoids have been found to reduce this inflammatory response. They inhibit the production of cytokines and inflammatory mediators, which positively impacts the condition and aids in recovery [71]. Several studies have indicated that combining antimicrobials with glucocorticoids significantly shortens the duration of fever, improves clinical symptoms, reduces complications, and prevents sequelae in severe and rapidly progressing RMPP cases [37,72,73].

This study has several limitations that should be considered when interpreting the findings. First, as a retrospective, single-center analysis conducted during a nationwide epidemic, the cases were selectively admitted to our institution, which may introduce selection bias and limit the generalizability of the conclusions. Second, the sample size, while substantial for a single center, is relatively limited, and validation through multi-regional, large-scale studies is warranted.

Most importantly, the initial treatment regimens were not standardized—a key limitation inherent to the real-world, observational design. In the context of the high macrolide-resistance epidemic of 2023–2024, clinicians frequently initiated alternative antibiotics (e.g., novel tetracyclines or fluoroquinolones) early for patients with more severe presentations or suspected resistance, rather than strictly observing a full 7-day macrolide course. Consequently, our non-RMPP group is more accurately a "clinically favorable outcome" cohort, which includes both true macrolide responders and patients who improved after an early switch to effective non-macrolide therapy. This heterogeneity in clinical management likely attenuates the observed association between genotypic macrolide resistance and the clinical outcome of RMPP (defined as failure after ≥7 days of macrolides), as some patients with resistant strains may have been "rescued" by early therapy escalation and thus excluded from the RMPP group. This represents an unavoidable confounding factor in non-randomized controlled studies. Although we adjusted for some confounders through multivariate analysis, treatment selection itself may be influenced by various unmeasured factors such as the initial severity of the child's condition and the clinician's judgment, leaving potential residual confounding. Future prospective studies employing a standardized initial treatment protocol are needed to more precisely quantify this relationship.

## 5. Conclusion

This study revealed a high prevalence of macrolide resistance (76.6%) in pediatric MPP, which was associated with point mutations at the A2063G and/or A2064G loci. The development of RMPP was independently associated with three key factors: the presence of these macrolide-resistant mutations, prolonged fever duration, and elevated D-dimer levels. Clinically, a fever duration >6.5 days and a D-dimer level >201.50 ng/mL served as strong predictors for RMPP, with D-dimer exhibiting the highest discriminatory power (AUC = 0.924), underscoring the role of hypercoagulability in RMPP pathogenesis. However, the pathogenesis of RMPP is multifactorial, involving an intricate interplay of factors, particularly complex immune dysregulation that remains incompletely elucidated. Future studies should broaden the scope of genetic screening and systematically delineate the underlying immunological mechanisms to inform more precise clinical management.

## Acknowledgments

We thank Bullet Edits Limited for the linguistic editing and proofreading of the manuscript.

## Author contributions

**Data curation:** Wei Gao, Xiaoyu Xiong.

**Methodology:** Xuan Tang.

**Supervision:** AiMin Li.

**Writing – original draft:** Wen Li.

**Writing – review & editing:** AiMin Li.

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
