## [Decision Letter · Decision Letter 0]

9 Dec 2025

Dear Dr. Li,

Thank you for submitting your manuscript to PLOS ONE. After careful consideration, we feel that it has merit but does not fully meet PLOS ONE’s publication criteria as it currently stands. Therefore, we invite you to submit a revised version of the manuscript that addresses the points raised during the review process.

**ACADEMIC EDITOR'S COMMENTS:**

We look forward to receiving your revised manuscript.

Kind regards,

Benjamin M. Liu, MBBS, PhD, D(ABMM), MB(ASCP)

Academic Editor

PLOS One

Journal Requirements:

“This study was supported by the 2024 Jingzhou Science and Technology Plan Project (Guidance), Project No. 2024HD47”

Reviewers' comments:

Reviewer's Responses to Questions

**Comments to the Author**

1. Is the manuscript technically sound, and do the data support the conclusions?

Reviewer #1: Yes

Reviewer #2: Yes

Reviewer #3: No

2. Has the statistical analysis been performed appropriately and rigorously?

Reviewer #1: Yes

Reviewer #2: Yes

Reviewer #3: No

3. Have the authors made all data underlying the findings in their manuscript fully available?

Reviewer #1: No

Reviewer #2: Yes

Reviewer #3: Yes

4. Is the manuscript presented in an intelligible fashion and written in standard English?

Reviewer #1: Yes

Reviewer #2: Yes

Reviewer #3: Yes

Reviewer #1: The manuscript titled "Correlation analysis of Mycoplasma pneumoniae 23S rRNA gene mutation with refractory Mycoplasma pneumoniae pneumonia in children” is a clinical relevant retrospective study written in a well organized manner. It explores the relationship between mutations in 23S rRNA of Mycoplasma pneumoniae and refractory Mycoplasma pneumoniae pneumonia (RMPP) disease in children. Study makes significant impact in the given rising prevalence of macrolide-resistant M. pneumoniae in Asia, and the authors provide a comprehensive data analysis from 205 pediatric cases.

However, while the study demonstrates statistical correlations and potential clinical implications, several methodological, interpretive, and linguistic issues should be addressed before publication.

Major revision suggested before accepting the manuscript. Comments are attached in a separate file.

Reviewer #2: Authors reported the correlation between mutations in 23S rRNA and RMPP in Children. However, the results indicate that the authors re-established the facts already available in the literatures. No references quoted for the method used for the detection of mutations and details of ELISA etc in the Materials and methods section.

Reviewer #3: I congratulate the authors for analysing the uncommon Mycoplasma pneumoniae infections. There are very few studies exploring the topics. Here are some points that needs to be addressed to make study more robust.

1. The term correlation is not appropriate for the analysis of the categorical variables. It should be restricted to analysis of the continuous variables. The term association will be better here, especially in the title.

2. The methodology does not describe at least in brief, with citation of the methodology, how the various laborartory tests were done

a. How was the MP-DNA done ?

b. How were the aforementioned mutations detected? Was sequencing done?

c. How were the MP-IgM and Ab titers detected and by which kit or methods ?

d. Nucleic acids for seven respiratory pathogens – how was it tested, which samples and which kits?

e. The different laboratory tests like wbc, crp , ldh, d-dimer and esr and PCT were included at what time point after admission? Is it the first test soon after admission?

3. How were the variables selected for the multivariable regression analyses?

4. None of the continuous variables in the table 1 are expressed as medians with IQR. Were all the subsets of the continuous variables for RMPP and non RMPP really having normal distribution? It seems highly unlikely. This is also reflected in the variables like the durations and the PCT (eg 0.32 vs 0.23, p – 0.026) values among the 2 groups look very similar, but have significant p value. Kindly recheck the normality of distribution of the continuous variables for each subset of drug resistant and non drug resistant groups and perform the t test only for those where the give variable has normal distribution for each of drug resistant and non drug resistant groups. If the variables are non normal in any of the 2 subgroups, alternative tests of significance needs to be used.

5. The table 2 also needs to be checked similarly for normality of distribution of the continuous variables.

6. Are the antibiotic/hormonal therapy groups in the table 7 mutually exclusive?

a. Why were the antibiotics other than macrolides used in the non RMPP groups?

b. Does it not interfere with the grouping in the study, where the RMPP group is defined when the patient worsens despite receiving 7 days of macrolide.

c. By this, is it not implicated that non- RMPP group will be one that recovers after < 7 days treatment with only macrolides?

7. Table 8:

a. How were the significant factors selected for the multiple logistic regression analysis.

b. What was the criteria used? Was any univariate regression analysis done for the selection of the significant factors to test for the multivariate analysis?

8. As mentioned in the results, 67.2% of the non RMPP had the drug resistant mutation.

a. How is this justified?

b. Does this not indicate that the detection of mutation is not a reliable test for detection of macrolide resistance?

c. Or is this because the groups used in the study are not proper.

d. Especially the non RMPP group, most probably includes patients that were not treated initially only by macrolides as evident in table 7.

e. If this is the case, the study is comparing the cases that did not respond to macrolides for seven days to a very heterogeneous group which received a variety of different initial antibiotics other than macrolides and hence the discrepancy in the mutation rates in the 2 groups. This not a good study design and the conclusions of the statististical analysis, hence, may not be reliable. The groups of RMPP and non-RMPP may need to be reviewed, which will affect the whole analysis and conclusions throughout the manuscript.

**Do you want your identity to be public for this peer review?** For information about this choice, including consent withdrawal, please see our Privacy Policy

Reviewer #1: **Yes:** Ananda Kumar Soshee

Reviewer #2: No

Reviewer #3: No

---

## [Author Response · Author response to Decision Letter 1]

14 Dec 2025

Please find my detailed point-by-point responses to all the reviewer and editor comments in the attached "Response to Reviewers" document.

---

## [Decision Letter · Decision Letter 1]

8 Jan 2026

Association analysis of Mycoplasma pneumoniae 23S rRNA gene mutation with refractory Mycoplasma pneumoniae pneumonia in children

PONE-D-25-51896R1

Dear Dr. Li,

We’re pleased to inform you that your manuscript has been judged scientifically suitable for publication and will be formally accepted for publication once it meets all outstanding technical requirements.

Kind regards,

Benjamin M. Liu, MBBS, PhD, D(ABMM), MB(ASCP)

Academic Editor

PLOS One

Additional Editor Comments (optional):

Reviewers' comments:

Reviewer's Responses to Questions

**Comments to the Author**

Reviewer #1: All comments have been addressed

Reviewer #3: All comments have been addressed

2. Is the manuscript technically sound, and do the data support the conclusions?

Reviewer #1: Yes

Reviewer #3: Yes

3. Has the statistical analysis been performed appropriately and rigorously?

Reviewer #1: Yes

Reviewer #3: Yes

4. Have the authors made all data underlying the findings in their manuscript fully available?

Reviewer #1: Yes

Reviewer #3: Yes

5. Is the manuscript presented in an intelligible fashion and written in standard English?

Reviewer #1: Yes

Reviewer #3: Yes

Reviewer #1: I found all necessary changes requested are incorporated into the manuscript. I recommend to accept the manuscript in the present form.

Thank you with warm regards.

Reviewer #3: The authors have satisfactorily addressed all the points raised in the previous review. The methodology is now clearly explained, the statistical analysis more robustly supports the conclusions, and the study’s limitations are clearly articulated. I commend the authors for their thorough and thoughtful response.

**Do you want your identity to be public for this peer review?** For information about this choice, including consent withdrawal, please see our Privacy Policy

Reviewer #1: No

Reviewer #3: No

---

## [Editor Report · Acceptance letter]

PONE-D-25-51896R1

PLOS One

Dear Dr. Li,

I'm pleased to inform you that your manuscript has been deemed suitable for publication in PLOS One. Congratulations! Your manuscript is now being handed over to our production team.

Kind regards,

on behalf of

Dr. Benjamin M. Liu

Academic Editor

PLOS One